# Cost-Sensitive Robustness against Adversarial Examples

**Xiao Zhang**
Department of Computer Science
University of Virginia
xz7bc@virginia.edu

**David Evans**
Department of Computer Science
University of Virginia
evans@virginia.edu

## Abstract

Several recent works have developed methods for training classifiers that are certifiably robust against norm-bounded adversarial perturbations. These methods assume that all the adversarial transformations are equally important, which is seldom the case in real-world applications. We advocate for *cost-sensitive robustness* as the criteria for measuring the classifier's performance for tasks where some adversarial transformation are more important than others. We encode the potential harm of each adversarial transformation in a cost matrix, and propose a general objective function to adapt the robust training method of Wong & Kolter (2018) to optimize for cost-sensitive robustness. Our experiments on simple MNIST and CIFAR10 models with a variety of cost matrices show that the proposed approach can produce models with substantially reduced cost-sensitive robust error, while maintaining classification accuracy.

## 1 Introduction

Despite the exceptional performance of deep neural networks (DNNs) on various machine learning tasks such as malware detection (Saxe & Berlin, 2015), face recognition (Parkhi et al., 2015) and autonomous driving (Bojarski et al., 2016), recent studies (Szegedy et al., 2014; Goodfellow et al., 2015) have shown that deep learning models are vulnerable to misclassifying inputs, known as *adversarial examples*, that are crafted with targeted but visually-imperceptible perturbations. While several defense mechanisms have been proposed and empirically demonstrated to be successful against existing particular attacks (Papernot et al., 2016; Goodfellow et al., 2015), new attacks (Carlini & Wagner, 2017; Tramèr et al., 2018; Athalye et al., 2018) are repeatedly found that circumvent such defenses. To end this arm race, recent works (Wong & Kolter, 2018; Raghunathan et al., 2018; Wong et al., 2018; Wang et al., 2018) propose methods to certify examples to be robust against some specific norm-bounded adversarial perturbations for given inputs and to train models to optimize for certifiable robustness.

However, all of the aforementioned methods aim at improving the *overall* robustness of the classifier. This means that the methods to improve robustness are designed to prevent seed examples in any class from being misclassified as any other class. Achieving such a goal (at least for some definitions of adversarial robustness) requires producing a perfect classifier, and has, unsurprisingly, remained elusive. Indeed, Mahloujifar et al. (2019) proved that if the metric probability space is concentrated, overall adversarial robustness is unattainable for any classifier with initial constant error.

We argue that overall robustness may not be the appropriate criteria for measuring system performance in security-sensitive applications, since only certain kinds of adversarial misclassifications pose meaningful threats that provide value for potential adversaries. Whereas overall robustness places equal emphasis on every adversarial transformation, from a security perspective, only certain transformations matter. As a simple example, misclassifying a malicious program as benign results in more severe consequences than the reverse.

In this paper, we propose a general method for adapting provable defenses against norm-bounded perturbations to take into account the potential harm of different adversarial class transformations. Inspired by cost-sensitive learning (Domingos, 1999; Elkan, 2001) for non-adversarial contexts, we capture the impact of different adversarial class transformations using a cost matrix $C$, where each

entry represents the cost of an adversary being able to take a natural example from the first class and perturb it so as to be misclassified by the model as the second class. Instead of reducing the overall robust error, our goal is to minimize the cost-weighted robust error (which we define for both binary and real-valued costs in $\boldsymbol{C}$). The proposed method incorporates the specified cost matrix into the training objective function, which encourages stronger robustness guarantees on cost-sensitive class transformations, while maintaining the overall classification accuracy on the original inputs.

**Contributions.** By encoding the consequences of different adversarial transformations into a cost matrix, we introduce the notion of *cost-sensitive robustness* (Section 3.1) as a metric to assess the expected performance of a classifier when facing adversarial examples. We propose an objective function for training a cost-sensitive robust classifier (Section 3.2). The proposed method is general in that it can incorporate any type of cost matrix, including both binary and real-valued. We demonstrate the effectiveness of the proposed cost-sensitive defense model for a variety of cost scenarios on two benchmark image classification datasets: MNIST (Section 4.1) and CIFAR10 (Section 4.2). Compared with the state-of-the-art overall robust defense model (Wong & Kolter, 2018), our model achieves significant improvements in cost-sensitive robustness for different tasks, while maintaining approximately the same classification accuracy on both datasets.

**Notation.** We use lower-case boldface letters such as $\boldsymbol{x}$ for vectors and capital boldface letters such as $\boldsymbol{A}$ to represent matrices. Let $[m]$ be the index set $\{1, 2, \ldots, m\}$ and $A_{ij}$ be the $(i, j)$-th entry of matrix $\boldsymbol{A}$. Denote the $i$-th natural basis vector, the all-ones vector and the identity matrix by $\boldsymbol{e}_i$, $\mathbf{1}$ and $\boldsymbol{I}$ respectively. For any vector $\boldsymbol{x} \in \mathbb{R}^d$, the $\ell_\infty$-norm of $\boldsymbol{x}$ is defined as $\|\boldsymbol{x}\|_\infty = \max_{i \in [d]} |x_i|$.

## 2 BACKGROUND

In this section, we provide a brief introduction on related topics, including neural network classifiers, adversarial examples, defenses with certified robustness, and cost-sensitive learning.

### 2.1 NEURAL NETWORK CLASSIFIERS

A $K$-layer neural network classifier can be represented by a function $f : \mathcal{X} \to \mathcal{Y}$ such that $f(\boldsymbol{x}) = f_{K-1}(f_{K-2}(\cdots(f_1(\boldsymbol{x}))))$, for any $\boldsymbol{x} \in \mathcal{X}$. For $k \in \{1, 2, \ldots, K-2\}$, the mapping function $f_k(\cdot)$ typically consists of two operations: an affine transformation (either matrix multiplication or convolution) and a nonlinear activation. In this paper, we consider rectified linear unit (ReLU) as the activation function. If denote the feature vector of the $k$-th layer as $\boldsymbol{z}_k$, then $f_k(\cdot)$ is defined as

$$\boldsymbol{z}_{k+1} = f_k(\boldsymbol{z}_k) = \max\{\boldsymbol{W}_k \boldsymbol{z}_k + \boldsymbol{b}_k, \mathbf{0}\}, \quad \forall k \in \{1, 2, \ldots K - 2\},$$

where $\boldsymbol{W}_k$ denotes the weight parameter matrix and $\boldsymbol{b}_k$ the bias vector. The output function $f_{K-1}(\cdot)$ maps the feature vector in the last hidden layer to the output space $\mathcal{Y}$ solely through matrix multiplication: $\boldsymbol{z}_K = f_{K-1}(\boldsymbol{z}_{K-1}) = \boldsymbol{W}_{K-1} \boldsymbol{z}_{K-1} + \boldsymbol{b}_{K-1}$, where $\boldsymbol{z}_K$ can be regarded as the estimated score vector of input $\boldsymbol{x}$ for different possible output classes. In the following discussions, we use $f_\theta$ to represent the neural network classifier, where $\theta = \{\boldsymbol{W}_1, \ldots, \boldsymbol{W}_{K-1}, \boldsymbol{b}_1, \ldots, \boldsymbol{b}_{K-1}\}$ denotes the model parameters.

To train the neural network, a loss function $\sum_{i=1}^N \mathcal{L}(f_\theta(\boldsymbol{x}_i), y_i)$ is defined for a set of training examples $\{\boldsymbol{x}_i, y_i\}_{i=1}^N$, where $\boldsymbol{x}_i$ is the $i$-th input vector and $y_i$ denotes its class label. Cross-entropy loss is typically used for multiclass image classification. With proper initialization, all model parameters are then updated iteratively using backpropagation. For any input example $\widetilde{\boldsymbol{x}}$, the predicted label $\widehat{y}$ is given by the index of the largest predicted score among all classes, $\operatorname{argmax}_j [f_\theta(\widetilde{\boldsymbol{x}})]_j$.

### 2.2 ADVERSARIAL EXAMPLES

An adversarial example is an input, generated by some adversary, which is visually indistinguishable from an example from the natural distribution, but is able to mislead the target classifier. Since "visually indistinguishable" depends on human perception, which is hard to define rigorously, we consider the most popular alternative: input examples with perturbations bounded in $\ell_\infty$-norm (Goodfellow et al., 2015). More formally, the set of adversarial examples with respect to seed example $\{\boldsymbol{x}_0, y_0\}$

and classifier $f_\theta(\cdot)$ is defined as

$$\mathcal{A}_\epsilon(\boldsymbol{x}_0, y_0; \theta) = \big\{ \boldsymbol{x} \in \mathcal{X} : \|\boldsymbol{x} - \boldsymbol{x}_0\|_\infty \leq \epsilon \text{ and } \operatorname*{argmax}_j [f_\theta(\boldsymbol{x})]_j \neq y_0 \big\}, \qquad (2.1)$$

where $\epsilon > 0$ denotes the maximum perturbation distance. Although $\ell_p$ distances are commonly used in adversarial examples research, they are not an adequate measure of perceptual similarity (Sharif et al., 2018) and other minimal geometric transformations can be used to find adversarial examples (Engstrom et al., 2017; Kanbak et al., 2018; Xiao et al., 2018). Nevertheless, there is considerable interest in improving robustness in this simple domain, and hope that as this research area matures we will find ways to apply results from studying simplified problems to more realistic ones.

## 2.3 DEFENSES WITH CERTIFIED ROBUSTNESS

A line of recent work has proposed defenses that are guaranteed to be robust against norm-bounded adversarial perturbations. Hein & Andriushchenko (2017) proved formal robustness guarantees against $\ell_2$-norm bounded perturbations for two-layer neural networks, and provided a training method based on a surrogate robust bound. Raghunathan et al. (2018) developed an approach based on semidefinite relaxation for training certified robust classifiers, but was limited to two-layer fully-connected networks. Our work builds most directly on Wong & Kolter (2018), which can be applied to deep ReLU-based networks and achieves the state-of-the-art certified robustness on MNIST dataset.

Following the definitions in Wong & Kolter (2018), an adversarial polytope $\mathcal{Z}_\epsilon(\boldsymbol{x})$ with respect to a given example $\boldsymbol{x}$ is defined as

$$\mathcal{Z}_\epsilon(\boldsymbol{x}) = \big\{ f_\theta(\boldsymbol{x} + \boldsymbol{\Delta}) : \|\boldsymbol{\Delta}\|_\infty \leq \epsilon \big\}, \qquad (2.2)$$

which contains all the possible output vectors for the given classifier $f_\theta$ by perturbing $\boldsymbol{x}$ within an $\ell_\infty$-norm ball with radius $\epsilon$. A seed example, $\{\boldsymbol{x}_0, y_0\}$, is said to be *certified robust* with respect to maximum perturbation distance $\epsilon$, if the corresponding adversarial example set $\mathcal{A}_\epsilon(\boldsymbol{x}_0, y_0; \theta)$ is empty. Equivalently, if we solve, for any output class $y_{\text{targ}} \neq y_0$, the optimization problem,

$$\operatorname*{minimize}_{\boldsymbol{z}_K} \; [\boldsymbol{z}_K]_{y_0} - [\boldsymbol{z}_K]_{y_{\text{targ}}}, \quad \text{subject to } \boldsymbol{z}_K \in \mathcal{Z}_\epsilon(\boldsymbol{x}_0), \qquad (2.3)$$

then according to the definition of $\mathcal{A}_\epsilon(\boldsymbol{x}_0, y_0; \theta)$ in (2.1), $\{\boldsymbol{x}_0, y_0\}$ is guaranteed to be robust provided that the optimal objective value of (2.3) is positive for every output class. To train a robust model on a given dataset $\{\boldsymbol{x}_i, y_i\}_{i=1}^N$, the standard robust optimization aims to minimize the sample loss function on the worst-case locations through the following adversarial loss

$$\operatorname*{minimize}_\theta \sum_{i=1}^N \max_{\|\boldsymbol{\Delta}\|_\infty \leq \epsilon} \mathcal{L}\big( f_\theta(\boldsymbol{x}_i + \boldsymbol{\Delta}), y_i \big), \qquad (2.4)$$

where $\mathcal{L}(\cdot, \cdot)$ denotes the cross-entropy loss. However, due to the nonconvexity of the neural network classifier $f_\theta(\cdot)$ introduced by the nonlinear ReLU activation, both the adversarial polytope (2.2) and training objective (2.4) are highly nonconvex. In addition, solving optimization problem (2.3) for each pair of input example and output class is computationally intractable.

Instead of solving the optimization problem directly, Wong & Kolter (2018) proposed an alternative training objective function based on convex relaxation, which can be efficiently optimized through a dual network. Specifically, they relaxed $\mathcal{Z}_\epsilon(\boldsymbol{x})$ into a convex outer adversarial polytope $\widetilde{\mathcal{Z}}_\epsilon(\boldsymbol{x})$ by replacing the ReLU inequalities for each neuron $z = \max\{\widehat{z}, 0\}$ with a set of inequalities,

$$z \geq 0, \quad z \geq \widehat{z}, \quad -u\widehat{z} + (u - \ell)z \leq -u\ell, \qquad (2.5)$$

where $u, \ell$ denote the lower and upper bounds on the considered pre-ReLU activation.[1] Based on the relaxed outer bound $\widetilde{\mathcal{Z}}_\epsilon(\boldsymbol{x})$, they propose the following alternative optimization problem,

$$\operatorname*{minimize}_{\boldsymbol{z}_K} \; [\boldsymbol{z}_K]_{y_0} - [\boldsymbol{z}_K]_{y_{\text{targ}}}, \quad \text{subject to } \boldsymbol{z}_K \in \widetilde{\mathcal{Z}}_\epsilon(\boldsymbol{x}_0), \qquad (2.6)$$

which is in fact a linear program. Since $\mathcal{Z}_\epsilon(\boldsymbol{x}) \subseteq \widetilde{\mathcal{Z}}_\epsilon(\boldsymbol{x})$ for any $\boldsymbol{x} \in \mathcal{X}$, solving (2.6) for all output classes provides stronger robustness guarantees compared with (2.3), provided all the optimal

---

[1] The elementwise activation bounds can be computed efficiently using Algorithm 1 in Wong & Kolter (2018).

objective values are positive. In addition, they derived a guaranteed lower bound, denoted by $J_\epsilon\big(\boldsymbol{x}_0, g_\theta(\boldsymbol{e}_{y_0} - \boldsymbol{e}_{y_\text{targ}})\big)$, on the optimal objective value of Equation 2.6 using duality theory, where $g_\theta(\cdot)$ is a $K$-layer feedforward dual network (Theorem 1 in Wong & Kolter (2018)). Finally, according to the properties of cross-entropy loss, they minimize the following objective to train the robust model, which serves as an upper bound of the adversarial loss (2.4):

$$\underset{\theta}{\text{minimize}} \quad \frac{1}{N} \sum_{i=1}^{N} \mathcal{L}\bigg( - J_\epsilon\big(\boldsymbol{x}_i, g_\theta(\boldsymbol{e}_{y_i} \cdot \boldsymbol{1}^\top - \boldsymbol{I})\big), y_i \bigg), \tag{2.7}$$

where $g_\theta(\cdot)$ is regarded as a columnwise function when applied to a matrix. Although the proposed method in Wong & Kolter (2018) achieves certified robustness, its computational complexity is quadratic with the network size in the worst case so it only scales to small networks. Recently, Wong et al. (2018) extended the training procedure to scale to larger networks by using nonlinear random projections. However, if the network size allows for both methods, we observe a small decrease in performance using the training method provided in Wong et al. (2018). Therefore, we only use the approximation techniques for the experiments on CIFAR10 (§4.2), and use the less scalable method for the MNIST experiments (§4.1).

## 2.4 COST-SENSITIVE LEARNING

Cost-sensitive learning (Domingos, 1999; Elkan, 2001; Liu & Zhou, 2006) was proposed to deal with unequal misclassification costs and class imbalance problems commonly found in classification applications. The key observation is that cost-blind learning algorithms tend to overwhelm the major class, but the neglected minor class is often our primary interest. For example, in medical diagnosis misclassifying a rare cancerous lesion as benign is extremely costly. Various cost-sensitive learning algorithms (Kukar & Kononenko, 1998; Zadrozny et al., 2003; Zhou & Liu, 2010; Khan et al., 2018) have been proposed in literature, but only a few algorithms, limited to simple classifiers, considered adversarial settings.[2] Dalvi et al. (2004) studied the naive Bayes classifier for spam detection in the presence of a cost-sensitive adversary, and developed an adversary-aware classifier based on game theory. Asif et al. (2015) proposed a cost-sensitive robust minimax approach that hardens a linear discriminant classifier with robustness in the adversarial context. All of these methods are designed for simple linear classifiers, and cannot be directly extended to neural network classifiers. In addition, the robustness of their proposed classifier is only examined experimentally based on the performance against some specific adversary, so does not provide any notion of certified robustness. Recently, Dreossi et al. (2018) advocated for the idea of using application-level semantics in adversarial analysis, however, they didn't provide a formal method on how to train such classifier. Our work provides a practical training method that hardens neural network classifiers with certified cost-sensitive robustness against adversarial perturbations.

## 3 TRAINING A COST-SENSITIVE ROBUST CLASSIFIER

The approach introduced in Wong & Kolter (2018) penalizes all adversarial class transformations equally, even though the consequences of adversarial examples usually depends on the specific class transformations. Here, we provide a formal definition of cost-sensitive robustness (§3.1) and propose a general method for training cost-sensitive robust models (§3.2).

### 3.1 CERTIFIED COST-SENSITIVE ROBUSTNESS

Our approach uses a cost matrix $\boldsymbol{C}$ that encodes the cost (i.e., potential harm to model deployer) of different adversarial examples. First, we consider the case where there are $m$ classes and $\boldsymbol{C}$ is a $m \times m$ binary matrix with $C_{jj'} \in \{0, 1\}$. The value $C_{jj'}$ indicates whether we care about an adversary transforming a seed input in class $j$ into one recognized by the model as being in class $j'$. If the adversarial transformation $j \to j'$ matters, $C_{jj'} = 1$, otherwise $C_{jj'} = 0$. Let $\Omega_j = \{j' \in [m] : C_{jj'} \neq 0\}$ be the index set of output classes that induce cost with respect to

---

[2]Given the vulnerability of standard classifiers to adversarial examples, it is not surprising that standard cost-sensitive classifiers are also ineffective against adversaries. The experiments described in Appendix B supported this expectation.

input class $j$. For any $j \in [m]$, let $\delta_j = 0$ if $\Omega_j$ is an empty set, and $\delta_j = 1$ otherwise. We are only concerned with adversarial transformations from a seed class $j$ to target classes $j' \in \Omega_j$. For any example $\boldsymbol{x}$ in seed class $j$, $\boldsymbol{x}$ is said to be *certified cost-sensitive robust* if the lower bound $J_\epsilon(\boldsymbol{x}, g_\theta(\boldsymbol{e}_j - \boldsymbol{e}_{j'})) \geq 0$ for all $j' \in \Omega_j$. That is, no adversarial perturbations in an $\ell_\infty$-norm ball around $\boldsymbol{x}$ with radius $\epsilon$ can mislead the classifier to any target class in $\Omega_j$.

The *cost-sensitive robust error* on a dataset $\{\boldsymbol{x}_i, y_i\}_{i=1}^N$ is defined as the number of examples that are not guaranteed to be cost-sensitive robust over the number of non-zero cost candidate seed examples:

$$\textit{cost-sensitive robust error} = 1 - \frac{\#\{i \in [N] : J_\epsilon(\boldsymbol{x}_i, g_\theta(\boldsymbol{e}_{y_i} - \boldsymbol{e}_{j'})) \geq 0, \forall j' \in \Omega_{y_i}\}}{\sum_{j|\delta_j=1} N_j},$$

where $\#\mathcal{A}$ represents the cardinality of a set $\mathcal{A}$, and $N_j$ is the total number of examples in class $j$.

Next, we consider a more general case where $\boldsymbol{C}$ is a $m \times m$ real-valued cost matrix. Each entry of $\boldsymbol{C}$ is a non-negative real number, which represents the cost of the corresponding adversarial transformation. To take into account the different potential costs among adversarial examples, we measure the cost-sensitive robustness by the average certified cost of adversarial examples. The cost of an adversarial example $\boldsymbol{x}$ in class $j$ is defined as the sum of all $C_{jj'}$ such that $J_\epsilon(\boldsymbol{x}, g_\theta(\boldsymbol{e}_j - \boldsymbol{e}_{j'})) < 0$. Intuitively speaking, an adversarial example will induce more cost if it can be adversarially misclassified as more target classes with high cost. Accordingly, the *robust cost* is defined as the total cost of adversarial examples divided by the total number of valued seed examples:

$$\textit{robust cost} = \frac{\sum_{j|\delta_j=1} \sum_{i|y_i=j} \sum_{j'\in\Omega_j} C_{jj'} \cdot \mathbb{1}\left(J_\epsilon(\boldsymbol{x}_i, g_\theta(\boldsymbol{e}_j - \boldsymbol{e}_{j'})) < 0\right)}{\sum_{j|\delta_j=1} N_j}, \tag{3.1}$$

where $\mathbb{1}(\cdot)$ denotes the indicator function.

## 3.2 COST-SENSITIVE ROBUST OPTIMIZATION

Recall that our goal is to develop a classifier with certified cost-sensitive robustness, as defined in §3.1, while maintaining overall classification accuracy. According to the guaranteed lower bound, $J_\epsilon\left(\boldsymbol{x}_0, g_\theta(\boldsymbol{e}_{y_0} - \boldsymbol{e}_{y_{\text{targ}}})\right)$ on Equation 2.6 and inspired by the cost-sensitive CE loss (Khan et al., 2018), we propose the following robust optimization with respect to a neural network classifier $f_\theta$:

$$\underset{\theta}{\text{minimize}} \; \frac{1}{N} \sum_{i \in [N]} \mathcal{L}\big(f_\theta(\boldsymbol{x}_i), y_i\big)$$
$$+ \alpha \sum_{j \in [m]} \frac{\delta_j}{N_j} \sum_{i|y_i=j} \log\left(1 + \sum_{j'\in\Omega_j} C_{jj'} \cdot \exp\big(-J_\epsilon(\boldsymbol{x}_i, g_\theta(\boldsymbol{e}_j - \boldsymbol{e}_{j'}))\big)\right), \tag{3.2}$$

where $\alpha \geq 0$ denotes the regularization parameter. The first term in Equation 3.2 denotes the cross-entropy loss for standard classification, whereas the second term accounts for the cost-sensitive robustness. Compared with the overall robustness training objective function (2.7), we include a regularization parameter $\alpha$ to control the trade-off between classification accuracy on original inputs and adversarial robustness.

To provide cost-sensitivity, the loss function selectively penalizes the adversarial examples based on their cost. For binary cost matrixes, the regularization term penalizes every cost-sensitive adversarial example equally, but has no impact for instances where $C_{jj'} = 0$. For the real-valued costs, a larger value of $C_{jj'}$ increases the weight of the corresponding adversarial transformation in the training objective. This optimization problem (3.2) can be solved efficiently using gradient-based algorithms, such as stochastic gradient descent and ADAM (Kingma & Ba, 2015).

## 4 EXPERIMENTS

We evaluate the performance of our cost-sensitive robustness training method on models for two benchmark image classification datasets: MNIST (LeCun et al., 2010) and CIFAR10 (Krizhevsky & Hinton, 2009). We compare our results for various cost scenarios with overall robustness training

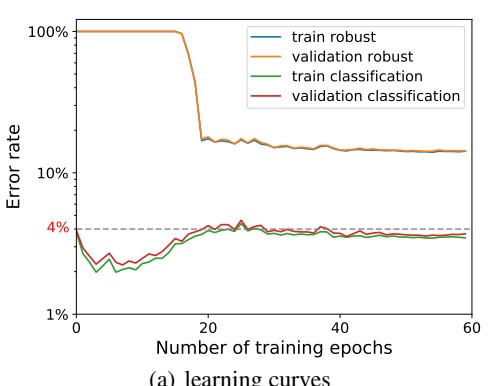
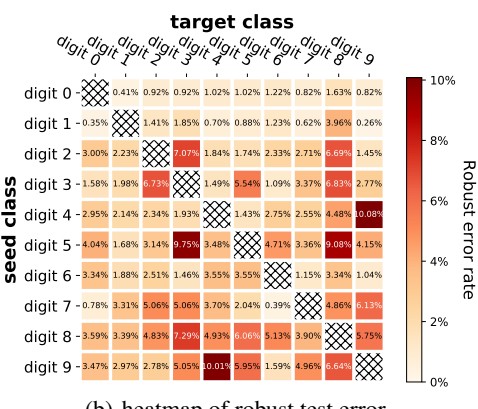

(a) learning curves      (b) heatmap of robust test error

Figure 1: Preliminary results on MNIST using overall robust classifier: (a) learning curves of the classification error and overall robust error over the 60 training epochs; (b) heatmap of the robust test error for pairwise class transformations based on the best trained classifier.

(§2.3) as a baseline. For both datasets, the relevant family of attacks is specified as all the adversarial perturbations that are bounded in an $\ell_\infty$-norm ball.

Our goal in the experiments is to evaluate how well a variety of different types of cost matrices can be supported. MNIST and CIFAR-10 are toy datasets, thus there are no obvious cost matrices that correspond to meaningful security applications for these datasets. Instead, we select representative tasks and design cost matrices to capture them.

### 4.1 MNIST

For MNIST, we use the same convolutional neural network architecture (LeCun et al., 1998) as Wong & Kolter (2018), which includes two convolutional layers, with 16 and 32 filters respectively, and a two fully-connected layers, consisting of 100 and 10 hidden units respectively. ReLU activations are applied to each layer except the last one. For both our cost-sensitive robust model and the overall robust model, we randomly split the 60,000 training samples into five folds of equal size, and train the classifier over 60 epochs on four of them using the Adam optimizer (Kingma & Ba, 2015) with batch size 50 and learning rate 0.001. We treat the remaining fold as a validation dataset for model selection. In addition, we use the $\epsilon$-scheduling and learning rate decay techniques, where we increase $\epsilon$ from 0.05 to the desired value linearly over the first 20 epochs and decay the learning rate by 0.5 every 10 epochs for the remaining epochs.

**Baseline: Overall Robustness.** Figure 1(a) illustrates the learning curves of both classification error and overall robust error during training based on robust loss (2.7) with maximum perturbation distance $\epsilon = 0.2$. The model with classification error less than 4% and minimum overall robust error on the validation dataset is selected over the 60 training epochs. The best classifier reaches 3.39% classification error and 13.80% overall robust error on the 10,000 MNIST testing samples. We report the robust test error for every adversarial transformation in Figure 1(b) (for the model without any robustness training all of the values are 100%). The $(i, j)$-th entry is a bound on the robustness of that seed-target transformation—the fraction of testing examples in class $i$ that cannot be certified robust against transformation into class $j$ for any $\epsilon$ norm-bounded attack. As shown in Figure 1(b), the vulnerability to adversarial transformations differs considerably among class pairs and appears correlated with perceptual similarity. For instance, only 0.26% of seeds in class 1 cannot be certified robust for target class 9 compare to 10% of seeds from class 9 into class 4.

**Binary Cost Matrix.** Next, we evaluate the effectiveness of cost-sensitive robustness training in producing models that are more robust for adversarial transformations designated as valuable. We consider four types of tasks defined by different binary cost matrices that capture different sets of adversarial transformations: *single pair*: particular seed class $s$ to particular target class $t$; *single seed*: particular seed class $s$ to any target class; *single target*: any seed class to particular target class $t$; and

Table 1: Comparisons between different robust defense models on MNIST dataset against $\ell_\infty$ norm-bounded adversarial perturbations with $\epsilon = 0.2$. The sparsity gives the number of non-zero entries in the cost matrix over the total number of possible adversarial transformations. The candidates column is the number of potential seed examples for each task.

| Task Description | | Sparsity | Candidates | Best $\alpha$ | Classification Error | | Robust Error | |
|---|---|---|---|---|---|---|---|---|
| | | | | | baseline | ours | baseline | ours |
| **single pair** | (0,2) | 1/90 | 980 | 10.0 | 3.39% | 2.68% | 0.92% | 0.31% |
| | (6,5) | 1/90 | 958 | 5.0 | 3.39% | 2.49% | 3.55% | 0.42% |
| | (4,9) | 1/90 | 982 | 4.0 | 3.39% | 3.00% | 10.08% | 1.02% |
| **single seed** | digit 0 | 9/90 | 980 | 10.0 | 3.39% | 3.48% | 3.67% | 0.92% |
| | digit 2 | 9/90 | 1032 | 1.0 | 3.39% | 2.91% | 14.34% | 3.68% |
| | digit 8 | 9/90 | 974 | 0.4 | 3.39% | 3.37% | 22.28% | 5.75% |
| **single target** | digit 1 | 9/90 | 8865 | 4.0 | 3.39% | 3.29% | 2.23% | 0.14% |
| | digit 5 | 9/90 | 9108 | 2.0 | 3.39% | 3.24% | 3.10% | 0.29% |
| | digit 8 | 9/90 | 9026 | 1.0 | 3.39% | 3.52% | 5.24% | 0.54% |
| **multiple** | top 10 | 10/90 | 6024 | 0.4 | 3.39% | 3.34% | 11.14% | 7.02% |
| | random 10 | 10/90 | 7028 | 0.4 | 3.39% | 3.18% | 5.01% | 2.18% |
| | odd digit | 45/90 | 5074 | 0.2 | 3.39% | 3.30% | 14.45% | 9.97% |
| | even digit | 45/90 | 4926 | 0.1 | 3.39% | 2.82% | 13.13% | 9.44% |

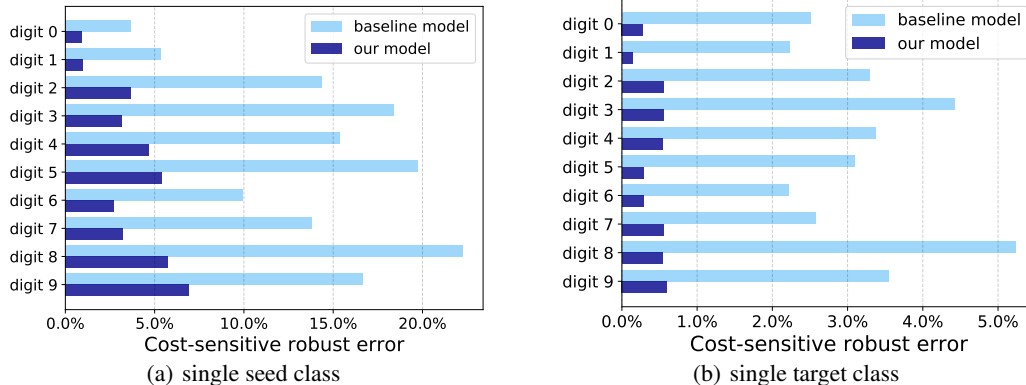

(a) single seed class      (b) single target class

Figure 2: Cost-sensitive robust error using the proposed model and baseline model on MNIST for different binary tasks: (a) treat each digit as the seed class of concern respectively; (b) treat each digit as the target class of concern respectively.

*multiple*: multiple seed and target classes. For each setting, the cost matrix is defined as $C_{ij} = 1$ if $(i, j)$ is selected; otherwise, $C_{ij} = 0$. In general, we expect that the sparser the cost matrix, the more opportunity there is for cost-sensitive training to improve cost-sensitive robustness over models trained for overall robustness.

For the single pair task, we selected three representative adversarial goals: a low vulnerability pair (0, 2), medium vulnerability pair (6, 5) and high vulnerability pair (4, 9). We selected these pairs by considering the robust error results on the overall-robustness trained model (Figure 1(b)) as a rough measure for transformation hardness. This is generally consistent with intuitions about the MNIST digit classes (e.g., 9 and 4 look similar, so are harder to induce robustness against adversarial transformation), as well as with the visualization results produced by dimension reduction techniques, such as t-SNE (Maaten & Hinton, 2008).

Similarly, for the single seed and single target tasks we select three representative examples representing low, medium, and high vulnerability to include in Table 1 and provide full results for all the single-seed and single target tasks for MNIST in Figure 2. For the multiple transformations

Table 2: Comparison results of different robust defense models for tasks with real-valued cost matrix.

| Dataset | Task | Sparsity | Candidates | Best $\alpha$ | Classification Error | | Robust Cost | |
|---------|------|----------|------------|---------------|----------|------|----------|------|
| | | | | | baseline | ours | baseline | ours |
| **MNIST** | small-large | 45/90 | 10000 | 0.04 | 3.39% | 3.47% | 2.245 | 0.947 |
| **MNIST** | large-small | 45/90 | 10000 | 0.04 | 3.39% | 3.13% | 3.344 | 1.549 |
| **CIFAR** | vehicle | 40/90 | 4000 | 0.1 | 31.80% | 26.19% | 4.183 | 3.095 |

task, we consider four variations: (i) the ten most vulnerable seed-target transformations; (ii) ten randomly-selected seed-target transformations; (iii) all the class transformations from odd digit seed to any other class; (iv) all the class transformations from even digit seed to any other class.

Table 1 summarizes the results, comparing the cost-sensitive robust error between the baseline model trained for overall robustness and a model trained using our cost-sensitive robust optimization. The cost-sensitive robust defense model is trained with $\epsilon = 0.2$ based on loss function (3.2) and the corresponding cost matrix $C$. The regularization parameter $\alpha$ is tuned via cross validation (see Appendix A for details). We report the selected best $\alpha$, classification error and cost-sensitive robust error on the testing dataset.

Our model achieves a substantial improvement on the cost-sensitive robustness compared with the baseline model on all of the considered tasks, with no significant increases in normal classification error. The cost-sensitive robust error reduction varies from 30% to 90%, and is generally higher for sparse cost matrices. In particular, our classifier reduces the number of cost-sensitive adversarial examples from 198 to 12 on the single target task with digit 1 as the target class.

**Real-valued Cost Matrices.** Loosely motivated by a check forging adversary who obtains value by changing the semantic interpretation of a number (Papernot et al., 2016), we consider two real-valued cost matrices: *small-large*, where only adversarial transformations from a smaller digit class to a larger one are valued, and the cost of valued-transformation is quadratic with the absolute difference between the seed and target class digits: $C_{ij} = (i - j)^2$ if $j > i$, otherwise $C_{ij} = 0$; *large-small*: only adversarial transformations from a larger digit class to a smaller one are valued: $C_{ij} = (i - j)^2$ if $i > j$, otherwise $C_{ij} = 0$. We tune $\alpha$ for the cost-sensitive robust model on the training MNIST dataset via cross validation, and set all the other parameters the same as in the binary case. The certified robust error for every adversarial transformation on MNIST testing dataset is shown in Figure 3, and the classification error and robust cost are given in Table 2. Compared with the model trained for overall robustness (Figure 1(b)), our trained classifier achieves stronger robustness guarantees on the adversarial transformations that induce costs, especially for those with larger costs.

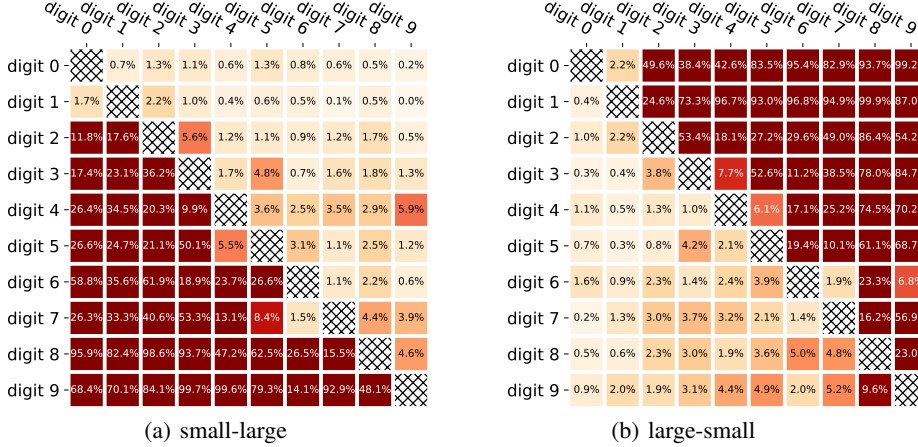

(a) small-large          (b) large-small

Figure 3: Heatmaps of robust test error using our cost-sensitive robust classifier on MNIST for various real-valued cost tasks: (a) *small-large*; (b) *large-small*.

Table 3: Cost-sensitive robust models for CIFAR10 dataset against adversarial examples, $\epsilon = 2/255$.

| Task Description | | Sparsity | Candidates | Best $\alpha$ | Classification Error | | Robust Error | |
|---|---|---|---|---|---|---|---|---|
| | | | | | baseline | ours | baseline | ours |
| **single pair** | (frog, bird) | 1/90 | 1000 | 10.0 | 31.80% | 27.88% | 19.90% | 1.20% |
| | (cat, plane) | 1/90 | 1000 | 10.0 | 31.80% | 28.63% | 9.30% | 2.60% |
| **single seed** | dog | 9/90 | 1000 | 0.2 | 31.80% | 30.69% | 57.20% | 28.90% |
| | truck | 9/90 | 1000 | 0.8 | 31.80% | 31.55% | 35.60% | 15.40% |
| **single target** | deer | 9/90 | 9000 | 0.1 | 31.80% | 26.69% | 16.99% | 3.77% |
| | ship | 9/90 | 9000 | 0.1 | 31.80% | 24.80% | 9.42% | 3.06% |
| **multiple** | A-V | 24/90 | 6000 | 0.1 | 31.80% | 26.65% | 16.67% | 7.42% |
| | V-A | 24/90 | 4000 | 0.2 | 31.80% | 27.60% | 12.07% | 8.00% |

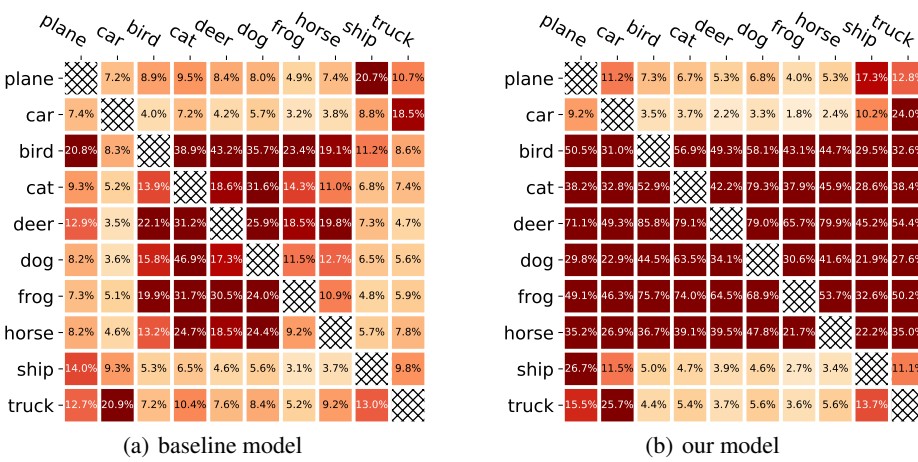

(a) baseline model            (b) our model

Figure 4: Heatmaps of robust test error for the real-valued task on CIFAR10 using different robust classifiers: (a) baseline model; (b) our proposed cost-sensitive robust model.

## 4.2 CIFAR10

We use the same neural network architecture for the CIFAR10 dataset as Wong et al. (2018), with four convolutional layers and two fully-connected layers. For memory and computational efficiency, we incorporate the approximation technique based on nonlinear random projection during the training phase (Wong et al. (2018), §3.2). We train both the baseline model and our model using random projection of 50 dimensions, and optimize the training objective using SGD. Other parameters such as learning rate and batch size are set as same as those in Wong et al. (2018).

Given a specific task, we train the cost-sensitive robust classifier on $80\%$ randomly-selected training examples, and tune the regularization parameter $\alpha$ according to the performance on the remaining examples as validation dataset. The tasks are similar to those for MNIST (§4.1), except for the multiple transformations task we cluster the ten CIFAR10 classes into two large groups: animals and vehicles, and consider the cases where only transformations between an animal class and a vehicle class are sensitive, and the converse.

Table 3 shows results on the testing data based on different robust defense models with $\epsilon = 2/255$. For all of the aforementioned tasks, our models substantially reduce the cost-sensitive robust error while keeping a lower classification error than the baseline.

For the real-valued task, we are concerned with adversarial transformations from seed examples in vehicle classes to other target classes. In addition, more cost is placed on transformations from vehicle to animal, which is 10 times larger compared with that from vehicle to vehicle. Figures 4(a)

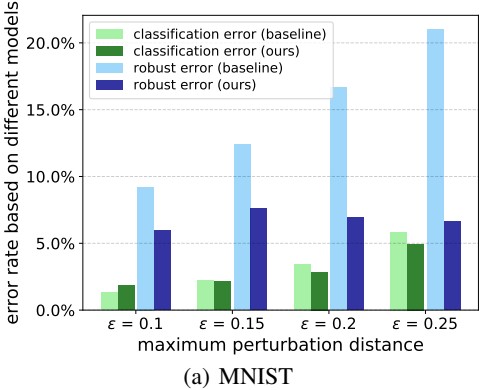 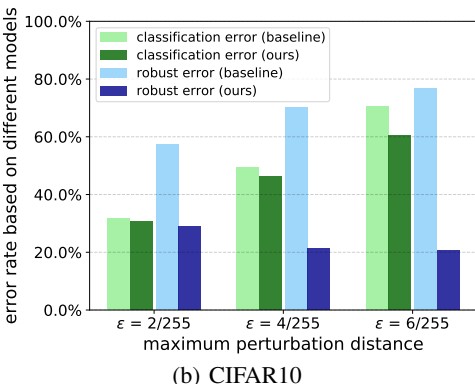

(a) MNIST

(b) CIFAR10

Figure 5: Results for different adversary strengths, $\epsilon$, for different settings: (a) MNIST single seed task with digit 9 as the chosen class; (b) CIFAR10 single seed task with dog as the chosen class.

and 4(b) illustrate the pairwise robust test error using overall robust model and the proposed classifier for the aforementioned real-valued task on CIFAR10.

### 4.3 Varying Adversary Strength

We investigate the performance of our model against different levels of adversarial strength by varying the value of $\epsilon$ that defines the $\ell_\infty$ ball available to the adversary. Figure 5 show the overall classification and cost-sensitive robust error of our best trained model, compared with the baseline model, on the MNIST single seed task with digit 9 and CIFAR single seed task with dog as the seed class of concern, as we vary the maximum $\ell_\infty$ perturbation distance.

Under all the considered attack models, the proposed classifier achieves better cost-sensitive adversarial robustness than the baseline, while maintaining similar classification accuracy on original data points. As the adversarial strength increases, the improvement for cost-sensitive robustness over overall robustness becomes more significant.

## 5 Conclusion

By focusing on overall robustness, previous robustness training methods expend a large fraction of the capacity of the network on unimportant transformations. We argue that for most scenarios, the actual harm caused by an adversarial transformation often varies depending on the seed and target class, so robust training methods should be designed to account for these differences. By incorporating a cost matrix into the training objective, we develop a general method for producing a cost-sensitive robust classifier. Our experimental results show that our cost-sensitive training method works across a variety of different types of cost matrices, so we believe it can be generalized to other cost matrix scenarios that would be found in realistic applications.

There remains a large gap between the small models and limited attacker capabilities for which we can achieve certifiable robustness, and the complex models and unconstrained attacks that may be important in practice. The scalability of our techniques is limited to the toy models and simple attack norms for which certifiable robustness is currently feasible, so considerable process is needed before they could be applied to realistic scenarios. However, we hope that considering cost-sensitive robustness instead of overall robustness is a step towards achieving more realistic robustness goals.

### Availability

Our implementation, including code for reproducing all our experiments, is available as open source code at https://github.com/xiaozhanguva/Cost-Sensitive-Robustness.

ACKNOWLEDGEMENTS

We thank Eric Wong for providing the implementation of certified robustness we built on, as well as for insightful discussions. We thank Jianfeng Chi for helpful advice on implementing our experiments. This work was supported by grants from the National Science Foundation (#1804603 and #1619098) and research awards from Amazon, Baidu, and Intel.

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

# Cost-Sensitive Robustness against Adversarial Examples
## Supplemental Materials

## A  PARAMETER TUNING

For experiments on the MNIST dataset, we first perform a coarse tuning on regularization parameter $\alpha$ with searching grid $\{10^{-2}, 10^{-1}, 10^0, 10^1, 10^2\}$, and select the most appropriate one, denoted by $\alpha_{coarse}$, with overall classification error less than $4\%$ and the lowest cost-sensitive robust error on validation dataset. Then, we further finely tune $\alpha$ from the range $\{2^{-3}, 2^{-2}, 2^{-1}, 2^0, 2^1, 2^2, 2^3\} \cdot \alpha_{coarse}$, and choose the best robust model according to the same criteria.

Figures 6(a) and 6(b) show the learning curves for task B with digit 9 as the selected seed class based on the proposed cost-sensitive robust model with varying $\alpha$ (we show digit 9 because it is one of the most vulnerable seed classes). The results suggest that as the value of $\alpha$ increases, the corresponding classifier will have a lower cost-sensitive robust error but a higher classification error, which is what we expect from the design of (3.2).

We observe similar trends for the learning curves for the other tasks, so do not present them here. For the CIFAR10 experiments, a similar tuning strategy is implemented. The only difference is that we use $35\%$ as the threshold of overall classification error for selecting the best $\alpha$.

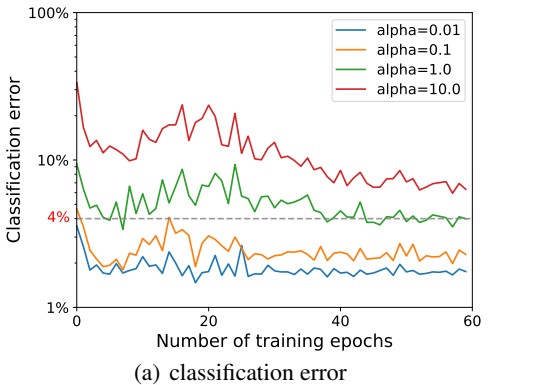
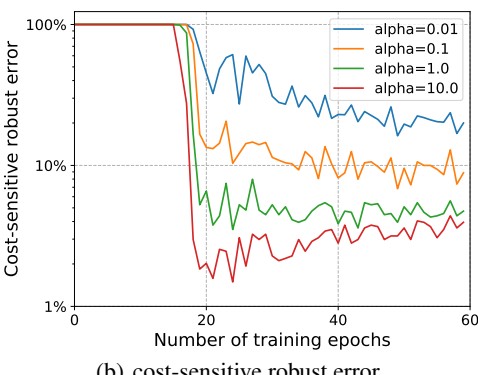

(a) classification error

(b) cost-sensitive robust error

Figure 6: Learning curves for single seed task with digit 9 as the selected seed class on MNIST using the proposed model with varying $\alpha$: (a) learning curves of classification error; (b) learning curves of cost-sensitive robust error. The maximum perturbation distance is specified as $\epsilon = 0.2$.

## B  COMPARISON WITH STANDARD COST-SENSITIVE CLASSIFIER

As discussed in Section 2.4, prior work on cost-sensitive learning mainly focuses on the non-adversarial setting. In this section, we investigate the robustness of the cross-entropy based cost-sensitive classifier proposed in Khan et al. (2018), and compare the performance of their classifier with our proposed cost-sensitive robust classifier. Given a set of training examples $\{(\mathbf{x}_i, y_i)\}_{i=1}^N$ and cost matrix $C$ with each entry representing the cost of the corresponding misclassification, the evaluation metric for cost-sensitive learning is defined as the average cost of misclassifications, or more concretely

$$\text{misclassification cost} = \frac{1}{N} \sum_{i \in [N]} C_{y_i \widehat{y}_i}, \quad \text{where } \widehat{y}_i = \operatorname*{argmax}_{j \in [m]} [f_\theta(\mathbf{x}_i)]_j,$$

where $m$ is the total number of class labels and $f_\theta(\cdot)$ denotes the neural network classifier as introduced in Section 2.1. In addition, the cross-entropy based cost-sensitive training objective takes the following form:

$$\operatorname*{minimize}_\theta \frac{1}{N} \sum_{j \in [m]} \sum_{i|_{y_i=j}} \log\left(1 + \sum_{j' \neq j} C_{jj'} \cdot \exp\left([f_\theta(\mathbf{x}_i)]_{j'} - [f_\theta(\mathbf{x}_i)]_j\right)\right). \tag{B.1}$$

Table 4: Comparison results of different trained classifiers for small-large real-valued task on MNIST with maximum perturbation distance $\epsilon = 0.2$.

| Classifier | Classification Error | Misclassification Cost | Robust Cost |
|---|---|---|---|
| **Baseline** | 0.91% | 0.054 | 85.197 |
| **Cost-Sensitive Standard** | 2.57% | 0.016 | 85.197 |
| **Overall Robustness** | 3.49% | 0.252 | 1.982 |
| **Cost-Sensitive Robustness** | 3.38% | 0.060 | 0.915 |

To provide a fair comparison, we assume the cost matrix used for (B.1) coincides with the cost matrix used for cost-sensitive robust training (3.2) in our experiment, whereas they are unlikely to be the same for real security applications. For instance, misclassifying a benign program as malicious may still induce some cost in the non-adversarial setting, whereas the adversary may only benefit from transforming a malicious program into a benign one.

We consider the small-large real-valued task for MNIST, where the cost matrix $C$ is designed as $C_{ij} = 0.1$, if $i > j$; $C_{ij} = 0$, if $i = j$; $C_{ij} = (i - j)^2$, otherwise. Table 4 demonstrates the comparison results of different classifiers in such setting: the baseline standard deep learning classifier, a standard cost-sensitive classifier (Khan et al., 2018) trained using (B.1), classifier trained for overall robustness (Wong & Kolter, 2018) and our proposed classifier trained for cost-sensitive robustness. Compared with baseline, the standard cost-sensitive classifier indeed reduces the misclassification cost. But, it does not provide any improvement on the robust cost, as defined in (3.1). In sharp contrast, our robust training method significantly improves the cost-sensitive robustness.

