# OpenReview forum: "Cost-Sensitive Robustness against Adversarial Examples"
_ICLR.cc/2019/Conference_

### Official Review · AnonReviewer2 · 2018-11-02

**Rating:** 8
**Confidence:** 3

**Review:**

** review score incremented following discussion below **

Strengths:

Well written and clear paper
Intuition is strong: not all source-target class pairs are as beneficial to find adversarial examples for

Weaknesses:

Cost matrices choices feel a bit arbitrary in experiments
CIFAR experiments still use very small norm-balls

The submission builds on seminal work by Dalvi et al. (2004), which studied cost-sensitive adversaries in the context of spam detection. In particular, it extends the approach to certifiable robustness introduced by Wong and Kolter with a cost matrix that specifies for each pair of source-target classes whether the model should be robust to adversarial examples that are able to take an input from the source class to the target (or conversely whether these adversarial examples are of interest to an adversary).

While the presentation of the paper is overall of great quality, some elements from the certified robustness literature could be reminded in order to ensure that the paper is self-contained. For instance, it is unclear how the guaranteed lower bound is derived without reading prior work. Adding this information in the present submission would make it easier for the reader to follow not only Sections 3.1 and 3.2 but also the computations behind Figure 1.b.

The experiments results are clearly presented but some of the details of the experimental setup are not always justified. If you are able to clarify the following choices in your rebuttal, this would help revise my review. First, the choice of cost matrices feels a bit arbitrary and somewhat cyclical. For instance, binary cost matrices for MNIST are chosen according to results found in Figure 1.b, but then later the same bounds are used to evaluate the performance of the approach. Yet, adversarial incentives may not be directly correlated with the “hardness” of a source-target class pair as measured in Figure 1.b. The real-valued cost matrices are better justified in that respect. Second, would you be able to provide additional justification or analysis of the choice of the epsilon parameter for CIFAR-10? For MNIST, you were able to improve the epsilon parameter from epsilon=0.1 to epsilon=0.2 but for CIFAR-10 the epsilon parameter is identical to Wong et al. Does that indicate that the results presented in this paper do not scale beyond simple datasets like MNIST?

Minor comments:


P2: The definition of adversarial examples given in Section 2.2 is a bit too restrictive, and in particular only applies to the vision domain. Adversarial examples are usually described as any test input manipulated by an adversary to force a model to mispredict.
P3: typo in “optimzation”
P5: trade off -> trade-off
P8: the font used in Figure 2 is small and hard to read when printed.

---

> ### Author Response · Authors · 2018-11-15
> **Thank you for your positive and constructive comments**
>
> We hope the following explanations address your questions:
>
> 1. Regarding the choice of the cost matrices
> Our goal in the experiments was to evaluate how well a variety of different types of cost matrices can be supported. MNIST and CIFAR-10 are toy datasets, thus defining cost matrices corresponding to meaningful security applications for these datasets is difficult. Instead, we selected representative tasks and designed cost matrices to capture them. Our experimental results show the promise of the cost-sensitive training method works across a variety of different types of cost matrices, so we believe it can be generalized to other cost matrix scenarios that would be found in realistic applications.
>
> It is a good point that the cost matrices that were selected based on the robust error rates in Fig 1B are somewhat cyclical, but it does not invalidate our evaluation. We use the “hardness” of adversarial transformation between classes only for choosing representative cost matrices, and the robust error results on the overall-robustness trained model as a measure for transformation hardness. Further, the transformation hardness implied by the robust error heatmap is generally consistent with intuitions about the MNIST digit classes (e.g., “9” and “4” look similar so are harder to make robust to transformation), as well as  with the visualization results produced by dimensional reduction techniques, such as t-SNE [1].
>
> 2. Regarding the choice of epsilon for CIFAR-10
> In our CIFAR-10 experiments, we set epsilon=2/255, the same experimental setup as in [2]. Our proposed cost-sensitive robust classifier can be applied to larger epsilon for CIFAR-10 dataset, and similar improvements have been observed for different epsilon settings. In particular, we have run experiments on CIFAR-10 with epsilon varying from {2/255, 4/255, 6/255} for the single seed task. The comparison results are reported in Figure 5(b), added to the revised PDF. These results support the generalizability of our method to larger epsilon settings.
>
> [1] Maaten and Hinton, Visualizing Data using t-SNE. http://www.jmlr.org/papers/v9/vandermaaten08a.html
> [2] Wong, et al., Scaling Provable Adversarial Defenses. https://arxiv.org/abs/1805.12514

---

> > ### Comment · AnonReviewer2 · 2018-11-19
> > **re: explanations**
> >
> > Thank you for taking the time to write a response to my review.
> >
> > Regarding 1., the explanation does provide some useful context for the choice of cost matrices. Do you have an intuition as to whether adversarial incentives will always correlate with transformation hardness? In other words, could there exist settings where the adversary would benefit more from a change in class that is relatively easy to make (and hard to defend against) compared to other class pairs?
> >
> > Thank you for providing additional experimental results regarding 2.

---

> > > ### Author Response · Authors · 2018-11-20
> > > **Adversarial incentives and transformation hardness**
> > >
> > > We don’t see any intrinsic reason why the class transformation difficulty is correlated with adversarial value, but the actual value and difficulty should depend on the application. The results in Table 1 show that the cost-sensitive robustness can harden both “easy” (4->9, robust error reduces from 10.08% to 1.02%) and “hard” (0->2, robust error reduces from 0.92% to 0.38%) - the improvement is bigger for the “easy” transformation, but even after the cost-sensitive robustness hardening, it remains slightly “easier” than the “hard” transformation in the overall robust model.
> > >
> > > For the MNIST classes, there is no correlation between the adversarial value (in the toy check fraud motivation) and transformation difficulty, since adversarial value is directional and semantically different digits can look more similar than far apart ones. For a more realistic security application, it would be desirable to define the classes in such a way that the valuable adversarial transformations are also the hardest ones to achieve.

---

> > > > ### Comment · AnonReviewer2 · 2018-11-20
> > > > **re: Adversarial incentives and transformation hardness**
> > > >
> > > > Thank you, the reduced effectiveness of the approach in settings where adversarial incentives do not align with class-pair hardness is the limitation I was concerned about in my previous comment. It would be great to add some elements of this discussion to the paper because your comments would help readers better understand cost matrices, as well as their applicability.
> > > >
> > > > I will increase my review score by one to take into account the outcome of this discussion.

---

> > > > > ### Author Response · Authors · 2018-11-26
> > > > > **Thank you for your suggestion.**
> > > > >
> > > > > Thank you for your consideration, we have included our discussions on the choices of cost matrices in Appendix D.

---

### Official Review · AnonReviewer3 · 2018-11-03
**An incremental paper that straightforwardly applies cost-sensitive loss to robust adversarial learning.**

**Rating:** 5
**Confidence:** 3

**Review:**

The paper introduces a new concept of certified cost-sensitive robustness against adversarial attacks. A cost-sensitive robust optimization formulation is then proposed for deep adversarial learning. Experimental results on two benchmark datasets (MNIST, CIFAR-10) are reported to show the superiority of the proposed method to overall robustness method, both with binary and real-value cost matrices.

The idea of cost-sensitive adversarial deep learning is well motivated. The proposed method is clearly presented and the results are easy to access. My main concern is about the novelty of the approach which looks mostly incremental as a rather direct extension of the robust model (Wong & Kolter 2018) to cost-sensitive setting. Particularly, the duality lower-bound based loss function and its related training procedure are almost identical to those from (Wong & Kolter 2018), up to certain trivial modification to respect the pre-specified misclassification costs. The numerical results show some promise. However, as a practical paper, the current empirical study appears limited in data scale: I believe additional evaluation on more challenging data sets can be useful to better support the importance of approach.

Pros:

- The concept of certified cost-sensitive robustness is well motivated and clearly presented.

Cons:

-  The novelty of method is mostly incremental given the prior work of (Wong & Kolter 2018).
- Numerical results show some promise of cost-sensitive adversarial learning in the considered settings, but still not supportive enough to the importance of approach.

---

> ### Author Response · Authors · 2018-11-15
> **Novelty is cost-sensitive robustness**
>
> Thank you for your review. Please see our responses below.
>
> 1. Concern regarding the novelty
> The review correctly notes that the method we use to achieve cost-sensitive robustness is a straightforward extension to the training procedure in Wong & Kolter (2018). The novelty of our paper lies in the introduction of cost-sensitive robustness as a more appropriate criteria to measure classifier’s performance, and in showing experimentally that the cost-sensitive robust training procedure is effective. Previous robustness training methods were designed for overall robustness, which does not capture well the goals of adversaries in most realistic scenarios. We consider it an advantage that our method enables cost-sensitive robustness to be achieved with straightforward modifications to overall robustness training.
>
> 2. Limitation in data scale
> We agree with the reviewer that certified robustness methods, including our work, are a long way from scaling to interesting models. All previous work on certified adversarial defenses has been limited to simple models on small or medium sized datasets (e.g., [1-3] below), but there is growing awareness that non-certified defenses are unlikely to resist adaptive adversaries and strong interest in scaling these methods. The method we propose and evaluate for incorporating cost-sensitivity in robustness training is generic enough that we expect it will also work with most improvements to certifiable robustness training. So, even though our implementation is not immediately practical today, we believe our results are of scientific interest, and the methods we propose are likely to become practical as rapid progress continues in scaling certifiable defenses.
>
>
> [1] Wong and Kolter, Provable defenses against adversarial examples via the convex outer adversarial polytope. https://arxiv.org/abs/1711.00851
> [2] Raghunathan, et al., Certified Defenses against Adversarial Examples. https://arxiv.org/abs/1801.09344
> [3] Wong, et al., Scaling Provable Adversarial Defenses. https://arxiv.org/abs/1805.12514

---

> > ### Comment · AnonReviewer3 · 2018-11-22
> > **Review update**
> >
> > Thank you for providing the feedback to clarify my concerns on novelty and experiment. There is no denying that cost-sensitive adversarial learning is an interesting topic worth exploring. I appreciate the efforts the authors put to introducing and evaluating a cost-sensitive extension of a robust DL model by Wong & Kolter (2018). However, what I found most frustrating about this work is its strong A-plus-B flavor which I still don’t think can stand for a novel scientific paper. Moreover, the adversarial learning part of the current method builds largely on (Wong & Kolter 2018). This looks somewhat narrow given the fairly broad rage of paper title and claims. I suggest including one or two additional certified robust learning models (e.g., PGD by Madry et al., 2018) into the proposed framework to better justify the importance of cost-sensitive robust learning.

---

> > > ### Author Response · Authors · 2018-11-26
> > > **Thank you for suggestions, but there is a misunderstanding on Madry et al. (2018) being a certified robust learning model.**
> > >
> > > Madry et al., (2018) is based on robust training against adversarially generated images devised via PGD attacks, which is not targeted for certifiable robustness. Thus, investigation on how to make PGD-based robust training cost-sensitive is beyond the scope of our work.
> > >
> > > To the best of our knowledge, before the submission of our paper there are only two proposed certifiable robust training methods: one is Wong & Kolter (2018) and the other one is [1]. Compared with Wong & Kolter (2018), [1] is only applicable to neural networks with two layers, thus we focus our experiments on Wong & Kolter (2018) that is more general. Recently, [2] extends the method of [1] to arbitrary number of neural network layers, thus it would be interesting to study whether our approach is applicable to the robust model developed in [2].
> > >
> > > Reference:
> > > [1] Raghunathan, et al., Certified Defenses against Adversarial Examples. https://arxiv.org/abs/1801.09344
> > > [2] Raghunathan, et al., Semidefinite relaxations for certifying robustness to adversarial examples. https://arxiv.org/abs/1811.01057

---

### Official Review · AnonReviewer1 · 2018-11-06
**interesting initiative, ad-hoc model**

**Rating:** 5
**Confidence:** 4

**Review:**

The authors define the notion of cost-sensitive robustness, which measures the seriousness of adversarial attack with a cost matrix. The authors then plug the costs of adversarial attack into the objective of optimization to get a model that is (cost-sensitively) robust against adversarial attacks.

The initiative is novel and interesting. Considering the long history of cost-sensitive learning, the proposed model is rather ad-hoc for two reasons:

(1) It is not clear why the objective should take the form of (3.1). In particular, if using the logistic function as a surrogate for 0-1 loss, shouldn't the sum of cost be in front of "log"? If using the probability estimated from the network in a Meta-Cost guided sense, shouldn't the cost be multiplied by the probability estimate (like 1/(1+exp(...))) instead of the exp itself? The mysterious design of (3.1) makes no physical sense to me, or at least other designs used in previous cost-sensitive neural network models like

Chung et al., Cost-aware pre-training for multiclass cost-sensitive deep learning, IJCAI 2016
Zhou and Liu, Training cost-sensitive neural networks with methods addressing the class imbalance problem, TKDE 2006 (which is cited by the authors)

are not discussed nor compared.

Update: I thank the authors for providing updated information in the Appendix discussing about other alternatives. While I still think it worth comparing with other approaches (as it is still not clear whether Khan's approach is regarded as state-of-the-art for *general* cost-sensitive deep learning), I think the authors have sufficiently justified their choice.

(2) It is not clear why the perturbed example should take the cost-sensitive form, while the original examples shouldn't (as the original examples follow the original loss). Or alternatively, if we optimize the original examples by the cost-sensitive loss, would it naturally achieve some cost-sensitive robustness (as the model would naturally make it harder to make high-cost mistakes)? Those issues are yet to be studied.

Update: I thank the authors for providing additional experiments on this part.

---

> ### Author Response · Authors · 2018-11-08
> **Objective justification**
>
> Thank you for your review. Your comments about the model being ad hoc stem from a few misunderstandings, which we hope to clarify:
>
> 1. Justification of training objective (3.1)
> The design of (3.1) is not ad hoc, but follows from previous cost-sensitive learning work such as MetaCost, and is inspired by the cost-sensitive CE loss (see equation (10) of [1] for a detailed definition). To be specific, class probabilities for cost-sensitive CE loss are computed by multiplying the corresponding cost and then normalizing the result vector. As a result, transformations that induce larger cost will receive larger penalization by minimizing the cost-sensitive CE loss. We neglected to include this explanation in the paper, and will revise it to make this clear.
>
> For the first question, moving the sum of cost in front of “log” is unreasonable because the loss for each seed example will not be a negative log-likelihood term as in the case of cross-entropy. We can check the sanity of the objective by examining whether it reduces to standard CE loss if we set C = 1*1^\top-I. For the second question, we indeed multiply the probability estimates by the cost, but the result vector has to be normalized before plugging into the cross entropy loss. Thus, the sum of cost will appear in front of the “exp” term.
>
> 2. Comparison with other alternative designs
> The cost-sensitive neural network models you mentioned are only demonstrated to be effective in the non-adversarial settings, whereas we show that our proposed classifier is effective in the adversarial setting. Thus, comparing our method with theirs is not appropriate, since it is unclear whether such alternative cost-sensitive models can be adapted and remain effective in the adversarial setting. Even if they can be adapted, it is still not the main focus of our paper, as our main goal is to show that our proposed classifier achieves significant improvements in cost-sensitive robustness in comparison with models trained for overall robustness.
>
> 3.  Why are the original examples are not in cost-sensitive form?
> The training objective (3.1) is constructed for maximizing both cost-sensitive robustness and standard classification accuracy, and allows us to use the alpha hyperparameter to control the weighting between these goals. Thus, the first term in (3.1) doesn’t involve cost-sensitivity. We regard the standard classification accuracy as an important criteria for measuring classifier performance. Besides, the cost matrix for misclassification of original examples might be different from the cost matrix of adversarial transformations. For instance, misclassifying a benign program as malicious may still induce some cost in the non-adversarial setting, whereas the adversary may only benefit from transforming a malicious program into a benign one. In a scenario where the model is cost-sensitive regardless of adversaries, it could make sense to incorporate a cost-sensitive loss function as the first term also, but we have not explored this and are focused on the adversarial setting where cost-sensitivity is with respect to adversarial goals.
>
> 4. What if we only optimize the original examples by cost-sensitive loss
> Given the vulnerability of deep learning classifiers against adversarial examples, we highly doubt that if we only optimize the original training by the cost-sensitive loss it would achieve significant cost-sensitive robustness (this expectation is based on how poorly models trained with the goal of overall accuracy do at achieving overall robustness). To be more convincing, we are running an experiment to test the robustness of a standard cost-sensitive classifier and will post the results soon.
>
> Reference
> [1]. Khan, et al., Cost-Sensitive Learning of Deep Feature Representations from Imbalanced Data. https://arxiv.org/abs/1508.03422

---

> > ### Comment · AnonReviewer1 · 2018-11-21
> > **thanks for clarifying**
> >
> > Thanks to the authors for clarifying.
> >
> > For point 1, I understand why the authors believe that their (3.1) is not ad hoc. Nevertheless, the authors' answers actually justify that (3.1) is perhaps not well-compared with other alternatives.
> > (a) (3.1) does not look the same from the cost-sensitive CE loss in [1], so why using (3.1) instead of [1] is a question mark.
> > (b) [1] contains more loss than the cost-sensitive CE loss, and why using a variant from cost-sensitive CE loss is another question mark.
> > (c) Even if (3.1) is a variant of cost-sensitive CE loss in [1], it hasn't been cited in this paper anyway?
> > (d) Quoting the authors, "transformations that induce larger cost will receive larger penalization by minimizing the cost-sensitive CE loss", but there are many different functions that achieve the property, including many discussed in other papers. Why or why not choosing (3.1)?
> > (e) Maybe it is because the derivation in Section 3 is way too short. But I fail to see how the authors follow MetaCost to "multiply the probability estimates by the cost, but the result vector has to be normalized before plugging into the cross entropy loss" and get (3.1). More detailed derivations are needed.
> >
> > I respectfully disagree with the authors' point on 2, as (IMHO) the authors are not using a state-of-art cost-sensitive objective (MetaCost is clearly outdated as evidenced by dozens of papers, and even [1] is just for the imbalanced setting, not for general cost-sensitivity that the authors want to achieve). So the current paper is like "adversarial learning + some cost-sensitive objective" gets better performance in the cost-sensitive setting. But cost-sensitive learning is a field that has been studied for more than 20 years. Why should we stick to "some cost-sensitive objective" but not "good/state-of-art cost-sensitive objective" when introducing cost-sensitivity to the adversarial setting? At least I demand to see a complete literature review on the cost-sensitive side (for the non-adversarial setting) and see the reasoning of the authors on the techniques that they choose to introduce to the adversarial learning field.
> >
> > I can accept the authors explanations on points 3 and 4, but still feel that it can be good to see what happens if the original examples are also evaluated/optimized cost-sensitively.

---

> > > ### Author Response · Authors · 2018-11-26
> > > **Please read Appendix B.3 and Appendix C in the revised pdf**
> > >
> > > 1. Comparison with other alternatives
> > > (a) We have added an equivalent form of the cost-sensitive CE loss for standard classification as in (B.1) in Appendix. The derivation of (B.1) simply follows the definition of the cross entropy loss and the modified softmax outputs y_n as defined in (11) of [1]. Our robust classifier is basically applying the same techniques on the guaranteed robust bound to induce cost-sensitivity for the adversarial setting.
> > >
> > > (b) Indeed, [1] introduces other cost-sensitive loss including MSE loss and SVM hinge loss besides the cost-sensitive CE loss. However, they only evaluated the cost-sensitive CE loss in their experiments, as argued in [1] that CE loss usually performs best among the three loss functions for multiclass image classification. Thus, we consider cost-sensitive robust optimization based on CE loss as the most promising approach.
> > >
> > > (c) We have cited [1] in Section 3.2 of the main paper in the revised pdf.
> > >
> > > (d) Please refer to Appendix C for the discussions of existing related work on cost-sensitive learning with neural networks, and explanations on why choosing to incorporate cost information into the cross entropy loss, instead of other loss functions.
> > >
> > > (e) The proposed robust training objective adapts cost-sensitive CE loss to the adversarial setting, and the cost-sensitive CE loss is aligned with the idea of minimizing the Bayes risks (see equation (1) in MetaCost). More specifically, it is proved in Lemma 10 of [1] that the cost-sensitive CE loss is c-calibrated, or more concretely, there exists an inverse relationship between the optimal CNN output and the Bayes cost of the t-th class.
> > > Therefore, minimization of the cost-sensitive CE loss will lead to classifier that has risks closer to the optimal Bayes risks.
> > >
> > > 2. As requested, we have added in Appendix C to survey related works on cost-sensitive learning for non-adversarial settings and explain the reasoning behind the techniques we choose.
> > >
> > > 3. Quoting from the reviewer, “What happens if the original examples are also evaluated/optimized cost-sensitively”, if you are referring to robustness of standard cost-sensitive learning method, this is what the experiment in Appendix B.3 tests. The results show naive cost-sensitive learning does not lead to cost-sensitive robustness.
> > >
> > > Reference
> > > [1]. Khan, et al., Cost-Sensitive Learning of Deep Feature Representations from Imbalanced Data. https://arxiv.org/abs/1508.03422

---

> ### Author Response · Authors · 2018-11-15
> **Additional experiments regarding cost-sensitive learning**
>
> We’ve added an Appendix B.3 to the revised paper that addresses the question you raised about whether standard cost-sensitive loss trained on original examples would improve cost-sensitive robustness. The results from our experiments show that standard cost-sensitive loss does not result in a classifier with cost-sensitive robustness.

---

### Meta-Review · Area_Chair1 · 2018-12-15
**practical important variant of certifiably robust classification**

**Confidence:** 4
**Recommendation:** Accept (Poster)

**Metareview:**

This paper studies the notion of certified cost-sensitive robustness against adversarial examples, by building from the recent [Wong & Koller'18]. Its main contribution is to adapt the robust classification objective to a 'cost-sensitive' objective, that weights labelling errors according to their potential damage.
This paper received mixed reviews, with a clear champion and two skeptical reviewers. On the one hand, they all highlighted the clarity of the presentation and the relevance of the topic as strengths; on the other hand, they noted the relatively little novelty of the paper relative [W & K'18]. Reviewers also acknowledged the diligence of authors during the response phase. The AC mostly agrees with these assessments, and taking them all into consideration, he/she concludes that the potential practical benefits of cost-sensitive certified robustness outweight the limited scientific novelty. Therefore, he recommends acceptance as a poster.